# A Probabilistic Model for Crystal Growth Applied to Protein Deposition at the Microscale

**DOI:** 10.3390/ma12030479

**Published:** 2019-02-04

**Authors:** Vicente J. Bolos, Rafael Benitez, Aitziber Eleta-Lopez, Jose L. Toca-Herrera

**Affiliations:** 1Department Matemáticas para la Economía y la Empresa, Facultad de Economía, Universidad de Valencia, Avda. Tarongers s/n, 46022 Valencia, Spain; vicente.bolos@uv.es; 2Self-Assembly Group, CIC nanoGUNE, Tolosa Hiribidea 76, 20018 Donostia/San Sebastián, Spain; a.eleta@nanogune.eu; 3Department of Nanobiotechnology, Institute of Biophysics, University of Natural Resources and Life Sciences (BOKU-Wien), Muthgasse 11, 1190 Vienna, Austria

**Keywords:** 2D crystal growth, protein crystal nucleation, probabilistic growth model

## Abstract

A probabilistic discrete model for 2D protein crystal growth is presented. This model takes into account the available space and can describe growing processes of a different nature due to the versatility of its parameters, which gives the model great flexibility. The accuracy of the simulation is tested against a real recrystallization experiment, carried out with the bacterial protein SbpA from *Lysinibacillus sphaericus* CCM2177, showing high agreement between the proposed model and the actual images of the crystal growth. Finally, it is also discussed how the regularity of the interface (i.e., the curve that separates the crystal from the substrate) affects the evolution of the simulation.

## 1. Introduction

A crystal is a (three-dimensional) periodic arrangement of repeating “structural motifs,” which can be atoms, molecules, or ions [1]. Technically, the process in which a (small) crystal becomes larger is called crystal growth. Crystallization is commonly referred to the creation of nucleation points first followed by crystal growth. Crystal growth is important for the understanding of biomineralization, molecular diffusion and adsorption, or fractal formation [2,3,4]. Theoretical crystallization models distinguish between thermodynamic and kinetic conditions [5,6]. For problems concerning growth rate and growth from limited resources, the Avrami [7,8] and the Gompertz functions can be used, respectively, for data modeling [9].

Crystal growth experiments can also be used to test growth and molecular adsorption models [10,11,12]. In this work we have taken advantage of a bacterial protein, the *Lysinibacillus sphaericus* surface layer protein precursor (sbpA) gene, from now on SbpA, which forms the outermost cell envelope of prokaryotic organisms (i.e., the crystalline bacterial cell surface layer) [13]. In particular, SbpA is able to self-assemble from solution, building square lattices on silicon dioxide and self-assembly monolayers [14]. Former studies on SbpA crystallization indicate on one hand that the process consists of a transition from an amorphous to a crystalline state [15] and, on the other hand, that a protein substrate determines the properties of the crystalline domain [16].

In this work we propose an approach to describe and model 2D protein crystal growth at the microscale. This implies that the model will simulate the spatial spread of the bulk crystal without considering the individual constituents of the crystal: their orientation, the nucleation process, and/or other energetic considerations between them and the substrate. The structure of the paper is as follows. In Section 2, we present the probabilistic model and define all the parameters involved in it. This model takes into account the available space for growing and reproduces different shapes and porosity (or lacunarity). In Section 3, we analyze the different parameters and how they affect the development of the simulation. All these different tunable parameters give the model a very high flexibility and allow us to successfully simulate real protein crystallization processes. Finally, in Section 4, we test the model by simulating the growth of an SbpA crystal. We conclude the paper discussing how the regularity of the interface affects the evolution of the crystallization process.

## 2. The Model

The model we present here is a discrete-space discrete-time model for a square region in which a regular n×n square mesh is defined. At each step k=1,2,…, one cell is filled by the crystal, and the corresponding time is denoted by $t_{k}$, with $t_{0}=0$ the initial time. In our model, $t_{k}$ is computed at each step *k* and determines only the growth rate of the crystal, but it does not affect the structure, i.e., its shape.

### 2.1. Structure

The space filled by the crystal when the *k*th cell is occupied is given by the *occupation matrix*
$M^{\left(k\right)}$, defined by $M_{ij}^{\left(k\right)}=1$ if cell (i,j) is occupied and by $M_{ij}^{\left(k\right)}=0$ if it is free.

In this way, we also define a *probability matrix*
$P^{\left(k\right)}$ so that the probability of the cell (i,j) to be occupied at the (k+1)th step is given by $P_{ij}^{\left(k\right)}$. Nevertheless, in the simulation procedure we shall use a *relative probability matrix*
$C^{\left(k\right)}$ which gives the number of chances that a cell has to be occupied, and it is related with $P^{\left(k\right)}$ in this way:$$P^{\left(k\right)}=\frac{C^{\left(k\right)}}{\sum_{i,j}C_{ij}^{\left(k\right)}}.$$

For example, if we want that each cell has exactly the same probability of being occupied at the beginning, then we can set $C_{ij}^{\left(0\right)}=1$. On the other hand, if we want the cells in a particular region to be twice as likely to be occupied at the beginning, then we can set $C_{ij}^{\left(0\right)}=2$ for such cells and $C_{ij}^{\left(0\right)}=1$ otherwise. Moreover, we can also have, for example, only one crystallization nucleus taking $C_{ij}^{\left(0\right)}=0$, except for one cell.

In our model, proteins will depose onto the substrate, occupying cells, following the next three structure rules:If a cell is occupied, it cannot be occupied again, i.e., if $M_{ij}^{\left(k\right)}=1$, then $C_{ij}^{\left(k\right)}=0$.If a cell is occupied, then the probability of occupation of the adjacent free cells is increased, see Equation (Equation 1).The probability of occupation of a free cell depends also on the available area in a neighborhood of that cell, see Equations (Equation 1) and (Equation 2).

This rules are independent of $t_{k}$ and hence they only determine the structure of the crystal, not its growth rate. Note that the first rule implies that this is a fully 2D model, i.e., no height increase is considered here.

Also note that, taking into account the second rule, the space scale is much larger than the characteristic length of the crystal structure. That is, each cell does not represent a single crystal, but a larger quantity. In fact, the scale is such that one occupied cell only increases the probability of occupation of the adjacent cells.

In our model, we propose that the chance of occupation of a free cell (i,j) at the *k*th step is given by
(1)$$C_{ij}^{\left(k\right)}=\left(C_{ij}^{\left(0\right)}+\rho \;\cdot \;{\left(A_{ij}^{\left(k\right)}\right)}^{\beta }\right)\;\cdot \;F_{ij}^{\left(k\right)}$$
where ρ is a positive constant, $A_{ij}^{\left(k\right)}$ is the number of adjacent occupied cells, and $0<F_{ij}^{\left(k\right)}\leq 1$ is a function related to the available area surrounding the free cell (see Equation (Equation 2)). The real parameter β determines the importance of the adjacent occupied cells and is usually set to 1 (see Section 3). Note that, if a free cell has no adjacent occupied cell, then its chance of occupation is $C_{ij}^{\left(k\right)}=C_{ij}^{\left(0\right)}\;\cdot \;F_{ij}^{\left(k\right)}$.

In order to speed up the algorithm, the third rule can be modified by only considering free cells adjacent to an occupied cell, so Equation (Equation 1) would only apply to free cells with $A_{ij}^{\left(k\right)}>0$. Hence, the chance of occupation of a free cell with no adjacent occupied cells (i.e., $A_{ij}^{\left(k\right)}=0$) would be $C_{ij}^{\left(k\right)}=C_{ij}^{\left(0\right)}$ instead of $C_{ij}^{\left(0\right)}\;\cdot \;F_{ij}^{\left(k\right)}$. It gives practically the same results when the number of crystallization nuclei is low and the algorithm is much faster.

For determining $A_{ij}^{\left(k\right)}$ we have to take into account that in the diagonal directions there is a weight factor of $\sqrt{2}/2$ (see Figure 1). This weight factor is set in order to avoid square-shaped evolution produced by the discretization in space, where circular ones should appear.

With respect to $F_{ij}^{\left(k\right)}$, first we have to define some concepts. The *effective radius*, $r_{\mathrm{eff}}$, of a free cell in a given direction with an *opening angle*
θ is the distance from the cell to the nearest occupied cell lying in the range of this direction. Then, the corresponding *effective area* is the area of the circular section with radius $r_{\mathrm{eff}}$ and angle θ (see Figure 2). The underlying idea is that the effective area is a zone where the free cell can capture proteins, but in this model, it is very simplified.

Although θ could be a parameter, in our model we take θ=π/2 because it greatly simplifies the discretized algorithm and it produces results matching the real measures (see Section 3). Moreover, we only take into account effective radii lesser than a given parameter called *influence radius*, $r_{\inf}$, beyond which, occupied cells do not hinder the crystal growth. Therefore, if $r_{\mathrm{eff}}>r_{\inf}$ (even $r_{\mathrm{eff}}=+\infty$), then we take $r_{\mathrm{eff}}=r_{\inf}$. Next, we define the *maximum effective radius*
$R_{ij}^{\left(k\right)}$ of a free cell (i,j) at the *k*th occupation stage as the maximum of the corresponding effective radius, but only in the eight main directions (for operational purposes): east, northeast, north, northwest, west, southwest, south and southeast.

Hence, for a free cell, we define
(2)$$F_{ij}^{\left(k\right)}=\left(1-\alpha \right)+\alpha {\left(\frac{R_{ij}^{\left(k\right)}}{r_{\inf}}\right)}^{\gamma }$$
where 0<α≤1, γ>0, and hence, $0<F_{ij}^{\left(k\right)}\leq 1$. The parameter α determines the importance of the maximum effective radius in the model and is usually taken close to 1. Setting α=0 means that the probability of occupation of a cell does not depend on the maximum effective radius and thus Rule 3 is not satisfied. On the other hand, we usually set γ=2 because in this case ${\left(R_{ij}^{\left(k\right)}/r_{\inf}\right)}^{2}$ is the ratio between the effective area and the maximum possible effective area (i.e., the area of the circular section of radius $r_{\inf}$ and angle θ). For convenience, $F_{ij}^{\left(k\right)}=0$ for an occupied cell. With this statement, Rule 1 is a consequence of Equation (Equation 1).

### 2.2. Kinetics

Let A(t) be the area defined by the crystal measured in occupied cells. Note that, although in this model A(t) is a step function, in this work we model the physical phenomena in which A(t) can be considered as a derivable real variable function. Nevertheless, we are going to consider the occupied area proportion
(3)$$y\left(t\right)=A\left(t\right)/n^{2}$$
that ranges from 0 to 1. In this section, we are going to set A(0)=0 and hence y(0)=0. Moreover, $A^{\prime }\left(0\right)$ will denote the number of expected occupied cells per time unit at t=0. It is not too important for the kinetics in a qualitative sense because, for example, a simulation with $A^{\prime }\left(0\right)=20$ will produce practically the same results as with $A^{\prime }\left(0\right)=10$ (leaving all other parameters unchanged), but in half the time.

With respect to the crystal growth rate, we can fit the the Avrami function, designed for modeling crystallization and some chemical reactions [7,8]:(4)$$y\left(t\right)=1-e^{-\kappa t^{\eta }}$$
where κ>0 and $\eta \in \left\{1,2,3,4\right\}$. However, in our case, where the crystal growth is two-dimensional, η can be considered equal to 1. Regarding κ, it is given by the initial crystal growth rate, since
(5)$$\kappa =A^{\prime }\left(0\right)/n^{2}.$$

The Avrami function fits well in the first stages of the crystal growth, but when the free space resource becomes scarce there are bigger differences (see Figure 3) since the Avrami function does not take into account this kind of resource. To solve this problem in the last stages, we can fit the Gompertz function, designed for growth with limited resources (in our case, the free space) [9]:(6)$$y\left(t\right)=e^{-be^{-ct}}$$
where b,c>0. However, note that this function can only model the crystal growth rate in an advanced state, since the initial occupied area in the Gompertz model is not zero. Therefore, a piecewise fit can be done using the Avrami function (Equation 4) for the first stages and the Gompertz function (Equation 6) for the rest, when the lack of free space becomes significant.

Finally, as an alternative for the piecewise fit, we can use a unique “hybrid” Avrami–Gompertz function:(7)$$y\left(t\right)=1-e^{-\kappa te^{-ct}},$$
where κ,c>0. Note that Equation (Equation 7) can only be applied in a time interval $\left[0,T\right]$ with T<1/c because, according to Equation (Equation 7), $y^{\prime }\left(t\right)<0$ for t>1/c. Moreover, the parameter κ in Equation (Equation 7) is also equal to $A^{\prime }\left(0\right)/n^{2}$, as in Equation (Equation 4), but the parameter *c* in Equation (Equation 7) is not the same as in Equation (Equation 6).

## 3. The Parameters

There are two kinds of parameters: *structural* and *kinetic*. The structural parameters determine the shape of the crystal, while the kinetic parameters are responsible for the crystal growth rate. We should also mention the parameter *n* (size of the square mesh) which is of great importance since its value affects both the structure and the kinetics of the crystal growth.

### 3.1. Structural Parameters

ρ: the “occupation chance multiplier” for free cells adjacent to an occupied one, see Equation (Equation 1). Mainly, it is responsible for the number of crystallization nuclei along the process (see Figure 4).θ: the opening angle, see Figure 2. In our model θ=π/2, so it is not taken as a parameter. Nevertheless, it can be changed, causing effects on the crystal shape: the greater the angle is, the bigger the “fjords” are (see Figure 5).β: the “sponginess” parameter, see Equation (Equation 1). Determines the effect of adjacent occupied cells on a free cell, and it is usually set to 1. In Figure 6, how this parameter (along with α) affects the shape of the crystal is shown.α: the “difficulty of filling” parameter, see Equation (Equation 2). It ranges from 0 to 1 and determines the importance of the maximum effective radius in the model. It is also related to the rate at which void regions are filled. Setting α=0 means that the probability of occupation of a cell does not depend on the maximum effective radius. In Figure 6 how this parameter (along with β) affects the shape of the crystal is shown.$r_{\inf}$: the influence radius, see Figure 2. It determines the range at which occupied cells hinder the crystal growth and thus the width of the “channels” (see Figure 7).γ: the “effective dimension” parameter, see Equation (Equation 2). It must be positive and, as explained before, in our model, is set to 2. It also determines how the maximum effective radius affects the crystal growth.

### 3.2. Kinetic Parameters

κ,η: parameters of the Avrami function in Equation (Equation 4) that models the first stages of the process. In our case, where the crystal growth is two-dimensional, it can be considered η=1. On the other hand, κ is determined by the initial crystal growth rate $A^{\prime }\left(0\right)$ (see Equation (Equation 5)).b,c: parameters of the Gompertz function in Equation (Equation 6) that models the last stages of the process.

## 4. Results

### 4.1. Protein Recrystallization

As was mentioned in the introduction, in order to validate the model, we tested it against the growth of the bacterial protein SbpA, which crystallizes in two dimensions forming structures called S-layers. This SbpA protein was recrystallized on an SiO${}_{2}$ substrate, and the overall process was scanned by an atomic force microscope (AFM) obtaining height images (see [14] for the specific details of the experiment). In Figure 8, the first row of images shows the crystallization process for times ranging from 10 to 110 min. The crystal growth continues steadily until more than 98% of the substrate is covered 12 h later.

Taking the 10 μm × 10 μm images of the crystal deposition at different time stamps, the total surface occupied by the crystal was estimated by transforming the original images to black and white 8-bit 530 × 530 pixel images and using a self-developed MATLAB code which computes the number of black pixels over the total amount of pixels of the image. It should be noted that each cell of the simulation process, which accounts for each pixel of the image, has a spatial resolution of about 18 nm × 18 nm, approximately. This gives us an accurate idea about the spatial validity of the model and how it describes the crystal evolution at the microscale. The second row of images in Figure 8 depicts, as an example, the transformation into black and white images of the original images shown in the top row.

Once the images are transformed and the occupied fraction is estimated, a growth curve is fit to the data according to the growth models described in Section 2.2 (see Figure 3). It is noteworthy how the Avrami model fits the data accurately at the beginning, but the growth rate for later moments is too high because it does not consider the limitations of space. As the crystal grows, the available space is reduced and so is the crystal growth rate. On the other hand, the Gompertz model captures the slow growth rate from the first 30 min, but it is not suitable for the first stages of the crystallization process. In fact, since the Gompertz growth model does not vanish at t=0, it cannot be applied there. Finally, the Avrami–Gompertz model is able to accurately fit the growth rate of the recrystallization process both at the beginning and at the final steady state moments. The non-linear least square fits were performed using the R statistical software [17] and the Levenberg–Marquardt algorithm provided by the minpack.LM package [18].

In order to assess the validity of the model as a description of the recrystallization process at the micro scale, we run several simulations using as an initial condition the black and white image of the crystallization at t=20 min. The choice of this initial condition is based on the fact that, with a 50% of occupied fraction, despite the stochastic nature of the model, the simulation would give results that could be compared to the experimental data. Figure 9 shows the evolution of the recrystallization process for one of the runs of the simulation (all of them gave results which were indistinguishable at a glance). The top row shows the black and white pictures obtained from the original AFM images while the second row depicts the simulation results at the same occupation fractions. The equivalence between the occupation fraction and the time stamp was obtained by the Avrami–Gompertz kinetic model. The initial (leftmost) pictures are the same in both rows. The close agreement between the simulation results and the experimental data is remarkable.

### 4.2. Influence of the Contour Regularity

Once the growth model has proven suitable for describing the protein recrystallization process, the question of the dependence of the growth rate on the regularity of the interface contour arises. The rationale behind all this is based on the fact that every system will evolve so its free energy at equilibrium is minimized. In crystal growth processes, the energy is highly dependent on the surface tension (or the line tension) for a new 2D interface [19]. Therefore, the larger the crystal contour length (determined by the crystal boundaries), the higher the free energy. Another way to understand this situation is to consider that a larger contour length implies more “unsaturated” or available binding sites for free protein in solution (analogous to surface tension in liquids). Thus, if the system seeks to minimize the free energy in the fastest possible way, the crystal should grow the most in the parts where the interface line is longer. Since the length of a curve is closely related to its regularity, we should find that a crystal with an irregular contour grows faster (under the same conditions) than a crystal with a regular one.

In order to assess how the regularity of the contour affects the growth process in our model, a simulation experiment was conducted with the following initial condition: a regular and an irregular nucleus both with the same size (measuring 10,000 occupied cells) were placed in opposite corners of a square big enough so the existence of any of the two nuclei did not affect the evolution of the other (see Figure 10 left picture). We run 2000 iterations of the simulation setting the parameters to β=1, α=0.5, and $r_{\inf}=50$. We also set $C_{ij}^{\left(0\right)}=0$ for cells with no adjacent occupied cells, so no new nuclei could appear. Figure 10 shows how the regular nucleus (bottom left corner) grows slower than the irregular nucleus (top right corner) and how its contour becomes less and less regular as the crystal deposition continues.

For a quantitative estimation of the differences between the regular and irregular nuclei, 100 simulations were conducted as described above for α=0.5 and α=0.8. Figure 11 shows how the growth of the regular nucleus is slower than the irregular one. For example, in the case of α=0.5, from the 2000 new cells grown by the simulation, only around 800 were added to the regular nucleus, while the other 1200 were added to the irregular one. It is also remarkable how, at the beginning, the differences in the growth rates are higher, and as the contour of the regular nucleus becomes more and more irregular, the growth rates are equalized. This also becomes clear in Figure 12, in which the ratio between the probabilities of occupation of a cell adjacent to both nuclei is plotted vs. the number of new deposited cells (ΔA). It can be seen how the curves are above 1, meaning that the irregular nucleus has more occupation probability than the regular one and how the curves decrease as the crystals grow, due to the fact that the differences in regularity between the two nuclei are less marked. In the long run, the quotients are asymptotically stabilized around a value slightly greater than 1, because the irregular nucleus has grown more in the initial stages of the simulation and these differences remain over time.

## 5. Discussion

The model presented in this work, based on simple probabilistic and geometrical rules, has proved to be very effective in simulating the spatial evolution of a protein crystal. The model is flexible enough to produce very different types of crystal shapes, ranging from dendritic to isotropic crystals, depending on the values of the structural parameters ρ, θ, β, and α (recall Figure 5 and Figure 6). Moreover, it is noteworthy that this model not only takes into account the current state of a crystal but also the space available for the crystal to grow depending on the values of the parameters $r_{\inf}$ and γ (Figure 7).

Other important feature of this model is the decoupling of the spatial evolution of the crystal from its change over time. The kinetics of the model is obtained separately from images obtained at different times and fitting the results to a mathematical growth curve. This allows us to determine the time stamp as a function of the crystallized area fraction. Regarding the growth curves used, we found that the modification of the Avrami model with the Gompertz growth function proved to be a good description of the evolution of the area occupied by the protein crystal over time.

It should be noted that very recently [20] other kinetic models have been developed. Those models are based on differential equations for population growth, and the functions involved are a combination of exponentials and hyperbolic tangents. However, such models predicted at the beginning slower growth rates than those obtained from our images; therefore, the fits were not better than those obtained with the Avrami–Gompertz function.

Finally, for the case of the simulation of the SpbA crystallization process, it is worth mentioning that, even though the core of the model algorithm is probabilistic, i.e., the determination of the cells that are going to be filled at a given time step is a certain random process depending only on the state of the crystal evolution at the previous step, once the available space for growing is small enough, the general evolution of the crystal ceases to be random and, from a macroscopic point of view (micro scale), becomes a deterministic process. In our case, when the model was initialized at t=20 min, when the occupied fraction was slightly higher than 50%, the model yielded practically the same results at each repetition of the simulation. An open question is if the proposed model could describe other adsorption/recrystallization processes leading to 2D regular structures (e.g., self-assembly monolayers and particle deposition).

## Figures and Tables

**Figure 1 materials-12-00479-f001:**
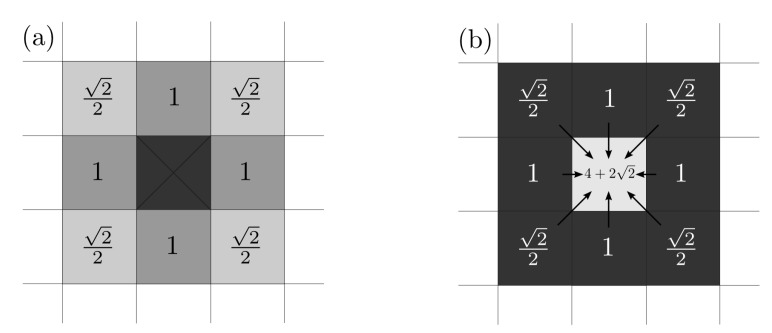
(**a**) Influence of an occupied cell (center) on the adjacent free cells. The weight in the diagonal directions is $\sqrt{2}/2$ for avoiding square-shaped evolutions. (**b**) For a fully surrounded free cell (center), its total number of occupied adjacent cells is $4+2\sqrt{2}$.

**Figure 2 materials-12-00479-f002:**
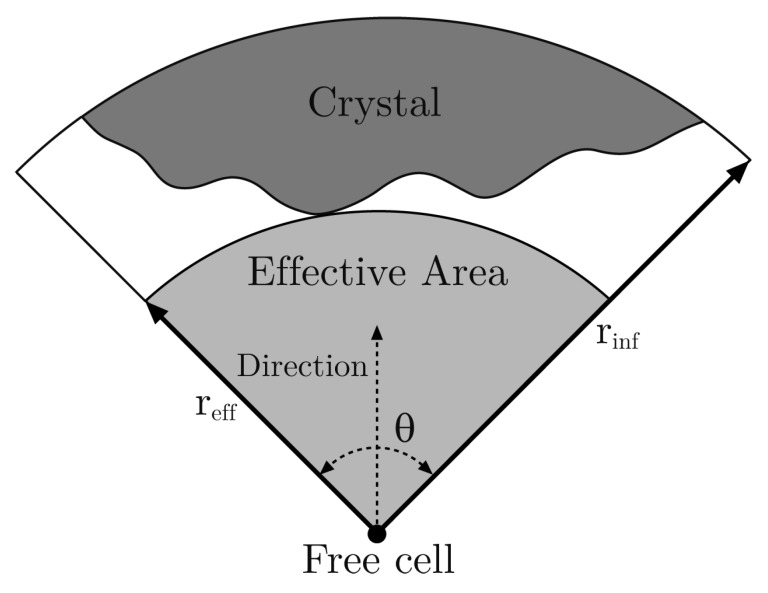
Effective area of a free cell in the north direction with an opening angle θ=π/2.

**Figure 3 materials-12-00479-f003:**
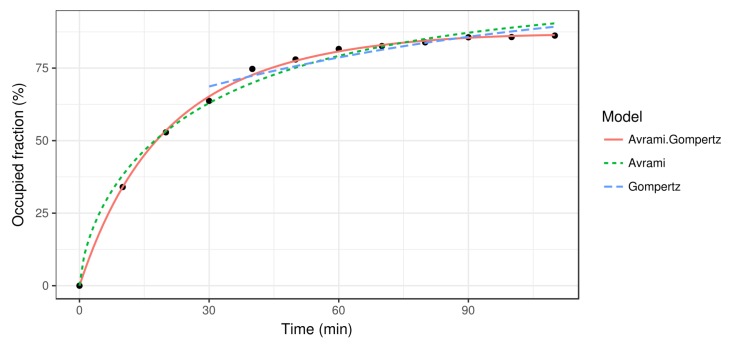
Estimation of the occupied fraction at different time stamps, from t=0 to t=120 min (black dots) together with the three different kinetic growth models: Avrami, Gompertz, and Avrami–Gompertz.

**Figure 4 materials-12-00479-f004:**
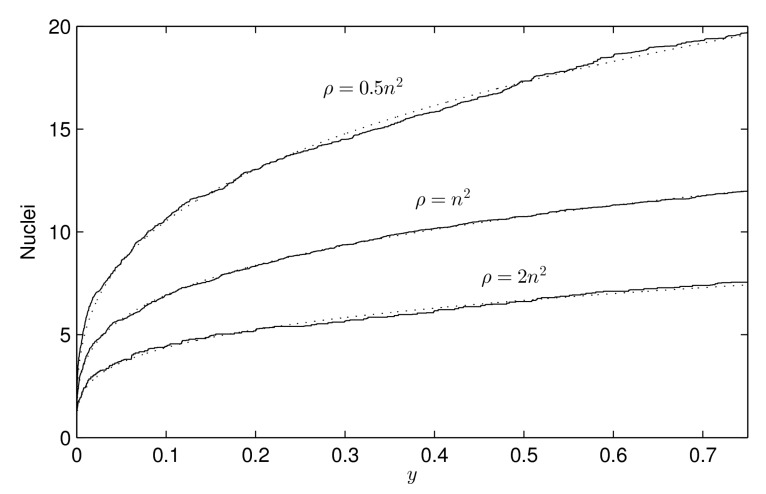
Number of crystallization nuclei along the process, from the beginning (y=0) to 75% of total occupied area (y=0.75), for different values of ρ. We represent (solid lines) the means of 100 simulations for $\rho =0.5n^{2}$, $\rho =n^{2}$, and $\rho =2n^{2}$ respectively, with n=256 and $C_{ij}^{\left(0\right)}=1$ for all cell (i,j). The rest of the parameters are β=1, α=0.8, and $r_{\inf}=32$, but they are not determinant. It is shown (dotted lines) that the number of crystallization nuclei follows a power law of the form $ay^{b}$ with a=21.41, b=0.3083 ($\rho =0.5n^{2}$), a=12.98, b=0.2722 ($\rho =n^{2}$), and a=8.006, b=0.2628 ($\rho =2n^{2}$). Note that parameter *a* represents the expected number of nuclei at the end of the process (y=1) and, from this result, it seems to hold $a_{1}\approx 1.635a_{2}$ for $\rho_{1}=\frac{1}{2}\rho_{2}$.

**Figure 5 materials-12-00479-f005:**
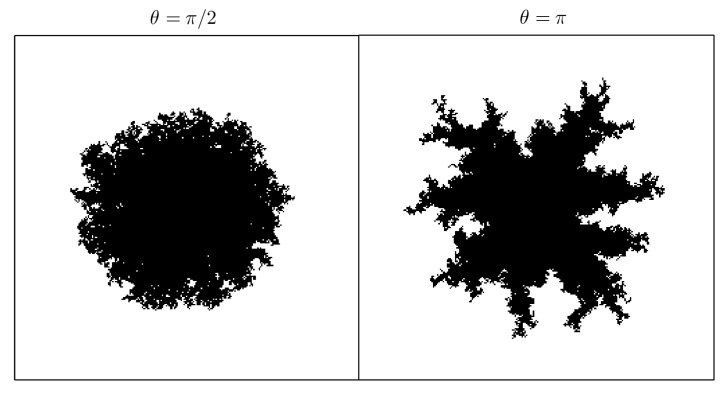
Shape at y=0.25 (i.e., 25% of total occupied area, see Equation (Equation 3)) of 1 crystallization nucleus for opening angle θ=π/2 and θ=π, taking α=0.8, β=1, $r_{\inf}=32$ and n=256. The greater the angle θ is, the bigger the “fjords” are. The simulations were carried out setting $C_{ij}^{\left(0\right)}=0$, except for one cell in the center.

**Figure 6 materials-12-00479-f006:**
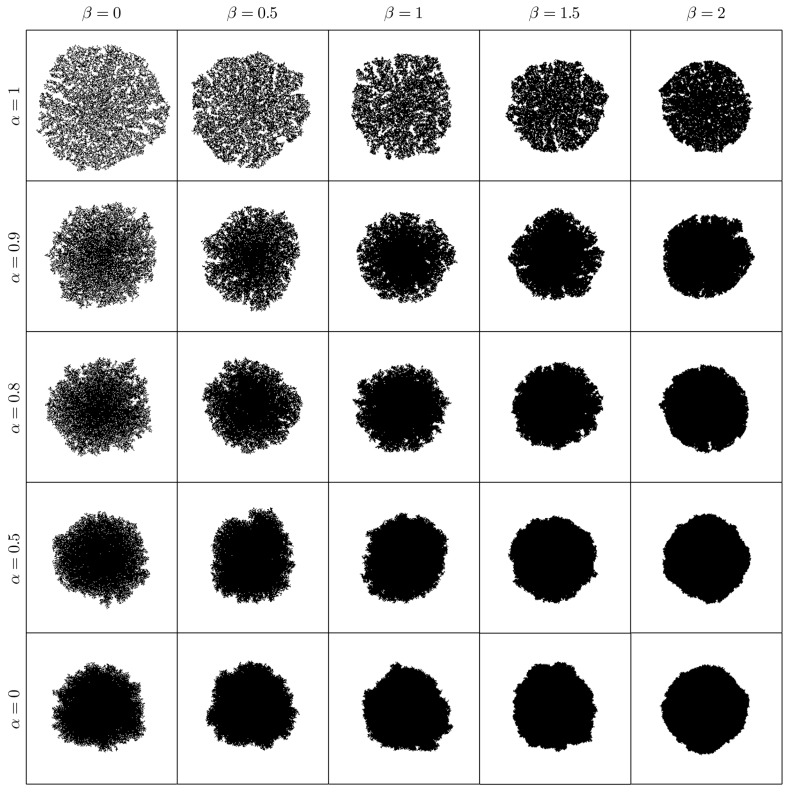
Shape at y=0.25 (i.e., 25% of total occupied area, see Equation (Equation 3)) of 1 crystallization nucleus for different values of α and β, taking n=256. The parameter β determines the “sponginess” of the crystal, and α represents the “difficulty of filling.” We have taken $r_{\inf}=32$, although it is not determinant in these figures. The simulations were carried out setting $C_{ij}^{\left(0\right)}=0$, except for one cell in the center.

**Figure 7 materials-12-00479-f007:**
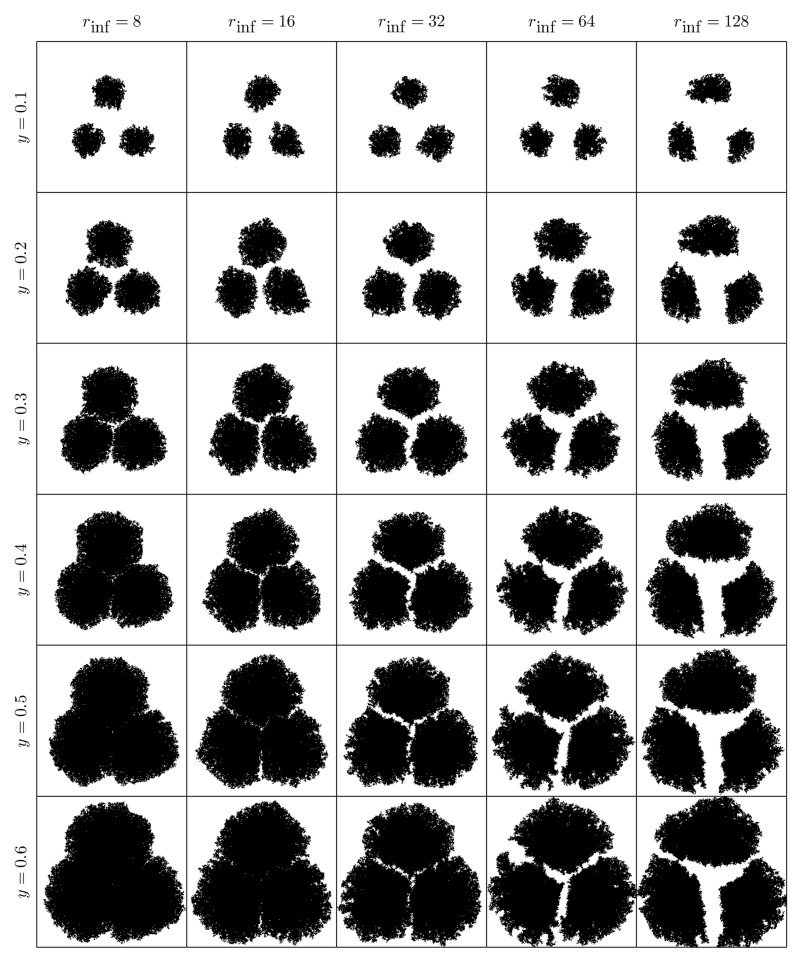
Evolution of 3 crystallization nuclei (for occupied area proportion from y=0.1 to y=0.6) with different values of $r_{\inf}$, taking α=0.8, β=1, and n=256. The parameter $r_{\inf}$ determines the width of the “channels”. The simulations were carried out setting $C_{ij}^{\left(0\right)}=0$, except for three cells placed at the vertices of a regular triangle.

**Figure 8 materials-12-00479-f008:**
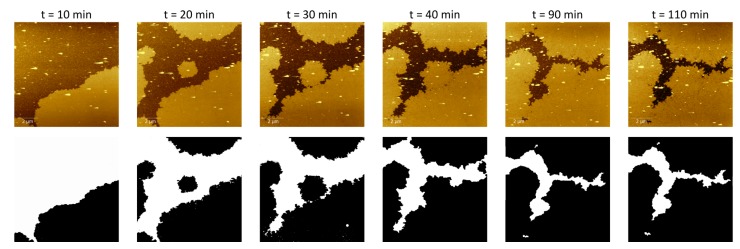
SbpA crystal growth sequence at different times. After the last time stamp at t=110 min, the crystal was stabilized, growing very slowly until the complete filling of the substrate more than 12 h after the beginning.

**Figure 9 materials-12-00479-f009:**
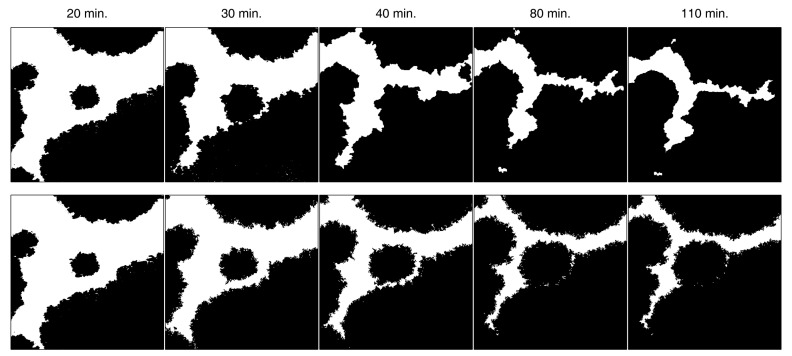
Comparison between real images (**up**) and simulation (**down**) at different times, with initial data corresponding to the real data at 20 min. The values of the structural parameters taken in the simulation are β=1, α=0.5, and $r_{\inf}=n/8$ (with n=256). The kinetic parameters have been set to fit the real data, according to the Avrami–Gompertz model (Equation 7).

**Figure 10 materials-12-00479-f010:**
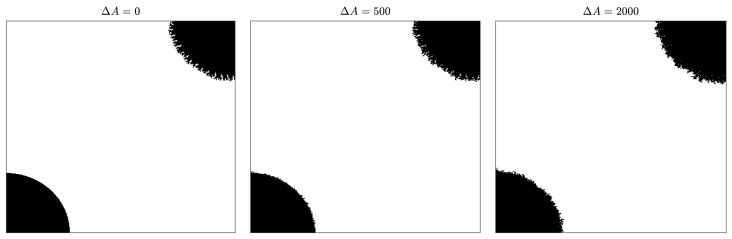
Evolution of two nuclei (irregular and regular), initially both with 10,000 occupied cells. The values of the structural parameters taken in the simulation are β=1, α=0.5, and $r_{\inf}=n/8$ (with n=400). We have taken $C_{ij}^{\left(0\right)}=0$ for cells with no adjacent occupied cells to prevent the formation of new nuclei.

**Figure 11 materials-12-00479-f011:**
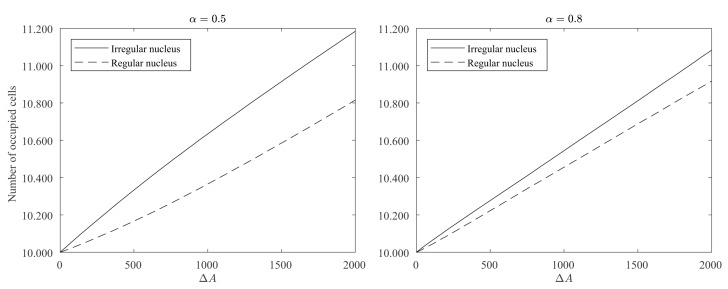
Evolution of the number of occupied cells for irregular and regular nuclei. We computed the mean of 100 simulations with the same parameter values as those used in the simulation of Figure 10 with α=0.5 and α=0.8.

**Figure 12 materials-12-00479-f012:**
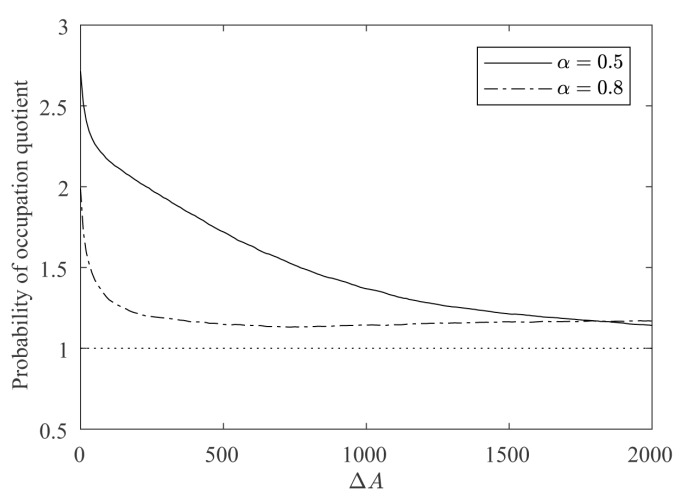
Evolution of the probability of occupation quotient, given by the probability of occupation of a cell adjacent to the irregular nucleus divided by the probability of occupation of a cell adjacent to the regular nucleus, with α=0.5 and α=0.8.

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
