# Peer review of "A Probabilistic Model for Crystal Growth Applied to Protein Deposition at the Microscale"

_materials, 2019, doi:10.3390/ma12030479_

Reviewer 1 Report

A discrete-space discrete-time model is suggested for simulations (using minpack.lm package) of crystal growth. The model is applied for studying crystallization processes occurring in a square region. Appropriate crystallization experiment (SbpA-recrystallization) is carried-out for comparison purposes; the reason being that SbpA crystalline arrays exhibit square lattice symmetry. It is shown that the Avrami-Gompertz model is able to accurately fit the growth rate of the SbpA-recrystallization, both at the beginning and at the final moments. The remarkable similarity between the experimental data and the simulation results (Fig. 9) shows that the growth model derived by the authors works excellently.

The calculation model is presented clearly. Sound rules serve as premises for the model; which main simplification is ϴ = π/2. There are no (evident) mistakes in the simulations. All aforementioned convinces me to recommend publication of the manuscript, but after some revision intended to render the paper more readily comprehensible for the broader readership of the Journal. My recommendation is to make more precise the following points:

- In the Introduction, the meaning of the abbreviation SbpA (a protein encoded by the gene sbpA, which forms the outermost cell envelope of prokaryotic organisms, i.e. the crystalline bacterial cell surface layer) must be given already on Line 19, where it is first mentioned – but not much latter, on lines 138-139. Also there, “the growth of a bacterial (?) which crystallizes” must be rewritten.

Wrong are the following statements:

- Lines 18-19: “Two dimensional in-situ protein crystallization is commonly utilized to study the function and structure of proteins [10].” In fact, single crystal X-ray diffraction is the universal, most powerful and accurate tool for biological macromolecule structure analysis and protein-substrate interactions. Ref. [10] describes in-vivo (not “in-situ”) protein crystallization, in which the crystals are not 2D.

- Lines 174-175: “The rationale behind all this is based on the fact that every system will evolve so its free energy is minimized.” This holds true for equilibrium shapes only. Crystal growth shapes are arising due to kinetic reasons - e.g., shapes resulting from diffusion-limited growth deviate from this tendency. An example is the dendritic crystal growth, e.g. the snow-flakes.

- Lines 177-178: “Thus if the system seeks to minimize the free energy in the fastest possible way, the crystal should grow the most in the parts were the interface line is longer.” In fact, attachment of crystal-building units is easier on rough surfaces, where these units are bond more strongly, with more neighbors (which is a kinetic factor). This gives a more sound explanation of the “regularity of the interface contour”-effect, which is described in Subsection 4.2.

Typos must be corrected, e.g.: Line 100: “the the Avrami function”.

Author Response

Reviewer #1:

Comments and Suggestions for Authors

[…] My recommendation is to make more precise the following points:

1. In the Introduction, the meaning of the abbreviation SbpA (a protein encoded by the gene sbpA, which forms the outermost cell envelope of prokaryotic organisms, i.e. the crystalline bacterial cell surface layer) must be given already on Line 19, where it is first mentioned – but not much latter, on lines 138-139. Also there, “the growth of a bacterial (?) which crystallizes” must be rewritten.

We thank the referee for the comment.

We have introduced the meaning of SbpA in the introduction of the revised manuscript. We have changed the sentence in line 20 to: “In this work we have taken advantage of a bacterial protein SbpA (a protein encoded by the gene sbpA) from Lysinibacillus sphaericus CCM2177, which forms the outermost cell envelope of prokaryotic organisms (i.e. the crystalline bacterial cell surface layer).”

Also we have changed lines 138-139 as follows: “As was mentioned in the introduction, in order to validate the model, we tested it against the growth of the bacterial protein SbpA, which crystallizes in two dimensions forming structures called S-layers.”

Wrong are the following statements:

2. - Lines 18-19: “Two dimensional in-situ protein crystallization is commonly utilized to study the function and structure of proteins [10].” In fact, single crystal X-ray diffraction is the universal, most powerful and accurate tool for biological macromolecule structure analysis and protein-substrate interactions. Ref. [10] describes in-vivo (not “in-situ”) protein crystallization, in which the crystals are not 2D.

We thank the referee. He/she is right. We have removed the reference and deleted the sentence.

3. - Lines 174-175: “The rationale behind all this is based on the fact that every system will evolve so its free energy is minimized.” This holds true for equilibrium shapes only. Crystal growth shapes are arising due to kinetic reasons - e.g., shapes resulting from diffusion-limited growth deviate from this tendency. An example is the dendritic crystal growth, e.g. the snow-flakes.

We thank the referee for the comment. We have changed the sentence “The rationale behind all this is based on the fact that every system will evolve so its free energy at equilibrium is minimized.”

4. Lines 177-178: “Thus if the system seeks to minimize the free energy in the fastest possible way, the crystal should grow the most in the parts were the interface line is longer.” In fact, attachment of crystal-building units is easier on rough surfaces, where these units are bond more strongly, with more neighbors (which is a kinetic factor). This gives a more sound explanation of the “regularity of the interface contour”-effect, which is described in Subsection 4.2.

We thank the reviewer for this particular comment.We believe that we share the same point of view as the reviewer. Our aim with this sentence is to give a plausible answer to the fact that our simulations tended to fill the voids near rough borders faster than near smooth boundaries. The first reason could indeed be the strength of the bonds between neighbors (the more neighbors the stronger the bond) as the reviewer points out. A plausible explanation for such kinetic factors is that once the process is finished the total (closed system) should have achieved a (local) minimum for the free energy (corresponding to an increase of entropy).

The idea would be that for rougher surfaces the interfacial energy might be larger, and by filling the voids it can be decreased. If the referee, has a more correct explanation we will be glad to include in the manuscript.

Typos must be corrected, e.g.: Line 100: “the the Avrami function”.

Corrected.

Reviewer 2 Report

The authors present a probabilistic discrete model for 2D protein crystal growth. They test the accuracy of their simulation against the crystallization behaviour observed for a s-layer protein.

First of all, the motivation for this work is not clear at all, what is the author’s goal to develop this model? i.e. what knowledge can the crystallization community gain from this research?

Secondly, although they claim that their model accurately reproduces the nucleation and growth of an s-layer protein, the data do not show these. It is not clear from the text how nucleation is modeled in this work? Accordingly, I can not see how it accurately reproduce the nucleation behavior of a real system. The data shown that surface fractals are formed, which has never been reported in literature for real systems. Quite the contrary, the crystallization of 2D protein layers reported in literature (e.g. Chung, Proceedings of the National Academy of Sciences 2010, 107, 16536–16541 et al., Sleutel et al., Nat. Commun., 5, 5598, 2014) shows compact nuclei that growth with rather smooth edges. Also, no voids are observed in real systems of 2D crystallites. What the authors present her resembles the outcome of a diffusion limited aggregation growth process, which stands very far from 2D growth of protein layers in natural systems (at least those that have been reported).

The role of the interactions of the protein molecules with the underlying surface is not discussed, but clearly this plays a important role in the nucleation and growth dynamics.

Author Response

Reviewer #2:

Comments and Suggestions for Authors

The authors present a probabilistic discrete model for 2D protein crystal growth. They test the accuracy of their simulation against the crystallization behaviour observed for a s-layer protein.

1. First of all, the motivation for this work is not clear at all, what is the author’s goal to develop this model? i.e. what knowledge can the crystallization community gain from this research?

The main objective of this is to develop a model allowing us to reproduce the kinetics of the protein crystals considered which is solely based on probabilistic and geometrical considerations.

2. Secondly, although they claim that their model accurately reproduces the nucleation and growth of an s-layer protein, the data do not show these. It is not clear from the text how nucleation is modeled in this work? Accordingly, I can not see how it accurately reproduce the nucleation behavior of a real system.

The reviewer is completely right and we thank him/her for the comment. We only mentioned the nucleation process in the abstract (because high resolution AFM measurements permit to visualize them). In fact, our simulation models the growth at micro-scale. The process of initialization of the crystal (nucleation) is simulated by a random initial process but it is not discussed here. We have remove the reference to nucleation from the abstract as we do not simulate the nucleation process from a prior stage, but we just considered the nucleation points as a randomly given initial condition for the simulation.

3. The data shown that surface fractals are formed, which has never been reported in literature for real systems. Quite the contrary, the crystallization of 2D protein layers reported in literature (e.g. Chung, Proceedings of the National Academy of Sciences 2010, 107, 16536–16541 et al., Sleutel et al., Nat. Commun., 5, 5598, 2014) shows compact nuclei that growth with rather smooth edges. Also, no voids are observed in real systems of 2D crystallites. What the authors present her resembles the outcome of a diffusion limited aggregation growth process, which stands very far from 2D growth of protein layers in natural systems (at least those that have been reported).

We believe there has been a misunderstanding, maybe because it was not fully clearly stated from our side, but our simulation process works at the micro-scale (the images are 10 um x 10 um). At such scale, the nucleation points are not distinguishable, and the line separating the crystal from the substrate is rough (fractal). This has been reported in the literature previously many times. Indeed, in both references given by the reviewer, the authors work at the nano-scale, but the images they give at larger scales are very similar to ours. For instance Figure 3 a) from Sleutel et al., Nat Commun., 5, 5598, 2014 and, is a picture 0.5 um x 0.5 um, we believe that this is quite similar to our images, and we might say that the line here is even rougher than ours. In Chung et al.,PNAS 2010, the largest images provided are of 1.5 um x 1.5 um (Figure 1 a and Figure 1 b) and the images where the crystal has grown to a comparable extent are of 50nm x 50nm and they do not look smooth to us (Figures 1e and 1 f). (see the pdf attached for the actual images).

4. The role of the interactions of the protein molecules with the underlying surface is not discussed, but clearly this plays a important role in the nucleation and growth dynamics.

As we pointed out above, the working scale considered in our simulations is of microns. This rules out the possibility of determining the interactions at the molecular level. We have given a simulation procedure that mimics the bulk crystal growth using some probabilistic and geometrical considerations. Obviously it would be extremely interesting to relate those probabilistic and geometrical parameters in our model to physical quantities related to interactions between the molecules and it is indeed a research line worth considering.

Reviewer 3 Report

The work of Bolos and co-workers attempt to simulate 2D protein crystal growth through a deposition process. Since I am not an expert in simulation, their results, particularly the good agreement between data and simulation of the growth of SbdA on SiO2 surfaces, catch my attention. Unfortunately, when reading carefully the experimental description of the simulation conditions it is impossible to find any energetic remark/reference for the SbdA crystallization/aggregation process (surface energy, etc. see Chernov, 2003), neither for anisotropic distribution of the surface interaction. In growing a crystal, whatever is 2D or 3D, anisotropic particles need to be properly oriented. I have not been able to find in the article where particle orientation is considered.

I understand that this is an approximation that seems to work but in its current format I cannot accept it for publication. I would like to ask the authors to try to explain better their approximations (Avrami and Gompertz) and why can it be used for protein crystal growth. Describe more in detail the protein growth unit and any spatial consideration token in the simulation. The article is written in a very physical oriented way, consider opening the scope to a wider range of field influence. Last, probably the example of in vivo protein crystallization (reference #10 in this manuscript) is not the best example to justify this work.

Suggested references:

Alexander A Chernov, Protein crystals and their growth, Journal of Structural Biology,142, 2003, 3-21,

DOI: 10.1016/S1047-8477(03)00034-0

J. Siódmiak, A. Gadomski, Growing lysozyme crystals under various physicochemical conditions: Computer modelling, Journal of Non-Crystalline Solids, 354, 2008, 4221-4226,

DOI: 10.1016/j.jnoncrysol.2008.06.084.

Andrzej M. Kierzek, Piotr Pokarowski, Piotr Zielenkiewicz, Microscopic model of protein crystal growth, Biophysical Chemistry, 87, Issue 1, 2000, 43-61,

DOI: 10.1016/S0301-4622(00)00179-4

S.D. Durbin, G. Feher, Simulation of lysozyme crystal growth by the Monte Carlo method, Journal of Crystal Growth, 110, 1991, 41-51,

https://doi.org/10.1016/0022-0248(91)90864-2

E.I. Givargizov, D.T.J. Hurle (Ed.), Handbook of Crystal Growth 3, Thin Film and Epitaxy, Part B: Growth Mechanics and Dynamics, Elsevier, Amsterdam (1994), p. 957

Author Response

Comments and Suggestions for Authors

1. The work of Bolos and co-workers attempt to simulate 2D protein crystal growth through a deposition process. Since I am not an expert in simulation, their results, particularly the good agreement between data and simulation of the growth of SbdA on SiO2 surfaces, catch my attention. Unfortunately, when reading carefully the experimental description of the simulation conditions it is impossible to find any energetic remark/reference for the SbdA crystallization/aggregation process (surface energy, etc. see Chernov, 2003), neither for anisotropic distribution of the surface interaction. In growing a crystal, whatever is 2D or 3D, anisotropic particles need to be properly oriented. I have not been able to find in the article where particle orientation is considered.

We thank the referee. The reviewer is right. We did not discuss particle orientation. The main reason is the scale of our model. It is a microscale model that tries to simulate, using simple rules, the spatial evolution of the bulk crystal. It does not account for individual particles and how they are positioned in space when forming the crystalline structure. We added a paragraph in the introduction so it is clearer for the reader the scale of the model: In lines 27-29: “In this work we propose an approach to describe and model 2D protein crystal growth at the microscale. This implies that the model will simulate the spatial spread of the bulk crystal without considering the individual constituents of the crystal: their orientation, the nucleation process and/or other energetic considerations between them and the substrate.”

2. I understand that this is an approximation that seems to work but in its current format I cannot accept it for publication. I would like to ask the authors to try to explain better their approximations (Avrami and Gompertz) and why can it be used for protein crystal growth.

We thank the referee for this particular critical comment.

The Avrami and Gompertz growth model functions and their hybridation in the Avrami –  Gompertz function were used in order to describe the evolution over time of the crystal extent. We are aware that the Avrami model relies on a differential equation relating to the rate of growth of the crystal to the number of individual crystal constituents present in it. However, we were only concerned with the description of the growth curve and, therefore, we look for typical mathematical functions describing growth with limitation of resources. Avrami and Gompertz are both examples of such functions as are as well other mathematical functions such as other sigmoid curves. For example, very recently, Stel et al. Chem. Commun., 2018, 54, 10264, used hyperbolic tangent functions combined with other exponential functions to describe the dynamics of the self-assembly of SbpA protein. In our case, for example a pure hyperbolic tangent also showed a good agreement with the measured data, although not as good as the Avrami-Gompertz function (see image in the attached pdf file).

3. Describe more in detail the protein growth unit and any spatial consideration token in the simulation. The article is written in a very physical oriented way, consider opening the scope to a wider range of field influence.

We have added a sentence explaining what the growth unit is as well as emphasizing the overall spatial scale of the simulation. In particular, in line 149 we added the size of the images (530x530 pixels) and in lines 150-153 we added the sentences: “It should be noted that each cell of the simulation process, which accounts for each pixel of the image, has a spatial resolution of about 18 x 18 nm, approximately. This gives us an accurate idea about the spatial validity of the model and how it describes the crystal evolution at the microscale.”

4. Last, probably the example of in vivo protein crystallization (reference #10 in this manuscript) is not the best example to justify this work.

The reviewer is right. We have removed this reference. We changed it for: “Crystal growth can also be used to test growth and molecular adsorption models (Chernov 2003, Siodmiak 2008 and Durbin 1991).”

Reviewer 4 Report

Comments for the authors:

This is an important study done by Bolos et al. Authors have done extensive study to understand the 2D protein crystal growth model, computationally as well experimentally. They have simulated the crystallization process computationally but also validated the prediction by doing real crystallization experiment using SbpA protein. Overall, this study is pretty unique in its own. Clear and unambiguous; and professional English used throughout in the article. Though the study has been carried out with great care and caution, however, some minor issues demand proper attention by the authors.

Comments.

1. Abstract, line 4: Please expand SbpA, as it is written first time.

2. Abstract, line2: ……its parameters which gives… Should be ……its parameters, which gives……

3. section 2.2, Kinetics, sentence above equation-4: we can fit the the Avrami function; Please delete one “the” as it is written two times.

4. Result page-6, line 140: Please expand AFM.

5. Author should include one more headline of “discussion” after the result section. In this, the important finding of the study should be discussed.

6. Please read the whole manuscript carefully once before resubmission, as I found minor grammatical error.

Author Response

Comments.

1. Abstract, line 4: Please expand SbpA, as it is written first time.

The meaning of spbA has been added in both the Abstract and the Introduction.

2. Abstract, line2: ……its parameters which gives… Should be ……its parameters, which gives……

Changed.

3. section 2.2, Kinetics, sentence above equation-4: we can fit the the Avrami function; Please delete one “the” as it is written two times.

Changed.

4. Result page-6, line 140: Please expand AFM.

Done. (Added “Atomic Force Microscope (AFM)”)

5. Author should include one more headline of “discussion” after the result section. In this, the important finding of the study should be discussed.

We thank the referee for this comment. A discussion section has been added.

6. Please read the whole manuscript carefully once before resubmission, as I found minor grammatical error.

Done.

Round  2

Reviewer 2 Report

In the revised version of their manuscript, the authors have made certain improvements. But, they have ignored other requests for improvement and the main problem of the work persists. The computer model presented, although scientifically sound, seems to have little bearing with how s-layers are formed around bacteria. Or at least the authors did not show any convincing experimental observations supporting their claim. As such the following statement (line 214-125): “The model presented in this work, based on simple probabilistic and geometrical rules, has proved to be very effective in simulating the spatial evolution of a protein crystal.” is not supported by the presented data. It is paramount that the authors include images of 2D protein crystal growth of s-layers, at the microscale, to support the results obtained from their model.

Moreover, in the worked cited by the authors “Chung, S.; Shin, S.H.; Bertozzi, C.R.; De Yoreo, J.J. Self-catalyzed growth of S layers via an 281 amorphous-to-crystalline transition limited by folding kinetics. Proceedings of the National Academy 282 of Sciences 2010, 107, 16536–16541. doi:10.1073/pnas.1008280107.” it is shown that first a conformation transition of the involved protein takes place, before crystallization. This surely most affect the kinetics of the growth process, and as such should also be considered in any model dealing with s-layer crystal growth. Moreover, other works have shown that background electrolytes also strongly affect the nucleation dynamics of 2D layers, how can this be accounted for in the model presented by the authors? (Clearly a microscopic model will not be able to model the molecular interactions per se, but the influence on microscopic crystal growth should be possible to model, since changes in the interactions at the molecular level are eventually relatable to macroscopic quantities such a solubility, supersaturation and interfacial energy).

Author Response

1. "In the revised version of their manuscript, the authors have made certain improvements. But, they have ignored other requests for improvement and the main problem of the work persists. The computer model presented, although scientifically sound, seems to have little bearing with how s-layers are formed around bacteria. Or at least the authors did not show any convincing experimental observations supporting their claim. As such the following statement (line 214-125): “The model presented in this work, based on simple probabilistic and geometrical rules, has proved to be very effective in simulating the spatial evolution of a protein crystal.” is not supported by the presented data. It is paramount that the authors include images of 2D protein crystal growth of s-layers, at the microscale, to support the results obtained from their model."

Answer: We think that there is a misunderstanding. Our work cannot reproduce the formation of “s-layers around bacteria”. No AFM in-situ experiment can do such thing. No 2-D S-layer crystallization can reproduce the conditions inside bacteria and neither was our objective.

The presented model deals with crystal growth at the microscale. That is why such results are shown. Our model does not take in to account nano-features of the protein crystal. However, AFM experiments done at the “nanoscale” revealed the typical crystalline pattern of the SbpA protein. We have investigated this protein since 2002. Eleta-Lopez and Toca-Herrera have shown in more than 20 publications the structural features of SbpA protein layers on different substrates (for different experimental conditions).

The substrate used in this work was (cleaned) silicon dioxide. This is the substrate that we used in the laboratory to test how the new batch of SbpA proteins do recrystallize. We have include in the supporting information a representative figure showing the lattice structure of the final crystal.

2. "Moreover, in the worked cited by the authors “Chung, S.; Shin, S.H.; Bertozzi, C.R.; De Yoreo, J.J. Self-catalyzed growth of S layers via an 281 amorphous-to-crystalline transition limited by folding kinetics. Proceedings of the National Academy 282 of Sciences 2010, 107, 16536–16541. doi:10.1073/pnas.1008280107.” it is shown that first a conformation transition of the involved protein takes place, before crystallization. This surely most affect the kinetics of the growth process, and as such should also be considered in any model dealing with s-layer crystal growth. Moreover, other works have shown that background electrolytes also strongly affect the nucleation dynamics of 2D layers, how can this be accounted for in the model presented by the authors? (Clearly a microscopic model will not be able to model the molecular interactions per se, but the influence on microscopic crystal growth should be possible to model, since changes in the interactions at the molecular level are eventually relatable to macroscopic quantities such a solubility, supersaturation and interfacial energy)."

Answer: We partially agree with the referee at this point. In the case of S-layers formed by the protein SbpA, crystal formation depends strongly form the presence of divalent cations in a basic solution (i.e. Ca2+, Mg2+). Former work carried out in our group have shown that monovalent and trivalent cations do not lead to the formation of crystalline protein layers. Another feature that is important for the formation of such protein crystal layers is the nature of the supporting substrate. If the substrate is hydrophilic less nucleation points are formed than on hydrophobic ones. Also for hydrophilic surfaces crystal formation takes longer (being the crystal domains larger). This is the reason why a macroscopic model can reproduce the growth of the protein crystal based on space availability (as a function of time). Our model is independent of the initial nucleation points and do not take into account molecular interactions. This is its strength. That is why we have propose it. It is not necessary to distinguish which (molecular) interaction drives the process.

Reviewer 3 Report

No further comments.

Author Response

Ok.

Thank you again for your useful insights.

The authors.